# Comparative Analysis of Biofilm Formation and Antibiotic Resistance in Five ESKAPE Pathogen Species from a Tertiary Hospital in Bangladesh

**DOI:** 10.3390/antibiotics14080842

**Published:** 2025-08-20

**Authors:** Tasnimul Arabi Anik, Rahat Uzzaman, Khandaker Toyabur Rahman, Abir Hossain, Faruk Islam, Mosammod Nowshin Tasnim, Shahin Ara Begum, Humaira Akhter, Anowara Begum

**Affiliations:** 1Environmental Microbiology Laboratory, Department of Microbiology, University of Dhaka, Dhaka 1000, Bangladesh; arabianik987@gmail.com (T.A.A.);; 2Laboratory Medicine, Green Life Hospital Ltd., Dhaka 1205, Bangladesh

**Keywords:** ESKAPE pathogens, AMR, biofilm formation, biofilm-related genes, carbapenemase

## Abstract

**Background**: Four of the six ESKAPE pathogens are responsible for a majority of antimicrobial resistance (AMR)-related deaths worldwide. Identifying the pathogens that evade antibiotic treatments more efficiently than others can help diagnose pathogens requiring more attention. The study was thus designed to evaluate the biofilm and resistance properties of five ESKAPE pathogens comparatively. A total of 165 clinical isolates of 5 ESKAPE pathogen species (*E. faecium*, *S. aureus*, *K. pneumoniae*, *A. baumannii*, and *P. aerurginosa*) were collected from a tertiary hospital in Bangladesh. **Methodology**: Following secondary identification, antibiotic susceptibility was determined by the disc diffusion method and minimum inhibitory concentration. The biofilm formation was determined by the microtiter plate biofilm formation assay. The biofilm-forming genes were screened by PCR. Detection of carbapenemase and Metallo-β-lactamase was performed by the modified carbapenem inactivation method (mCIM) and the EDTA-modified carbapenem inactivation method (eCIM) test, respectively. **Results**: Among Gram-positive isolates, *E. faecium* exhibited higher multi-drug resistance (MDR) rates (90%) compared to *S. aureus* (10%). In Gram-negative isolates, *A. baumannii* and *K. pneumoniae* showed elevated resistance to carbapenems (74.29% and 45.71%, respectively), cephalosporins, and β-lactam inhibitors, while *P. aeruginosa* demonstrated relatively lower resistance. Colistin resistance was highest in *K. pneumoniae* (42.86%). Biofilm formation was prevalent, with 88.5% of isolates forming biofilms, including 15.8% strong biofilm producers. Notably, *K. pneumoniae* and *A. baumannii* exhibited higher biofilm-forming capabilities compared to *P. aeruginosa*. A significant correlation was observed between biofilm formation and resistance to carbapenems, cephalosporins, and piperacillin/tazobactam (*p* < 0.05), suggesting a potential role of biofilms in disseminating resistance to these antibiotics. Carbapenemase production was detected in 23.8% of Gram-negative isolates, with *K. pneumoniae* showing the highest prevalence (34.3%). Additionally, 45.8% of carbapenemase producers expressed Metallo-β-lactamases (MBLs). Among *S. aureus* isolates, 46.7% carried the *mecA* gene, confirming methicillin resistance (MRSA), while 20% of *E. faecium* isolates exhibited vancomycin resistance, primarily mediated by the *vanB* gene. **Conclusions**: These findings can help pinpoint the pathogens of significant threat.

## 1. Introduction

While the number of bacteria resistant to multiple antibiotics continues to climb, the development of new antibiotics is slowing down, raising concerns about our ability to combat these infections in the future. The U.S. Centers for Disease Control and Prevention (CDC) and the World Health Organization (WHO) recognize antimicrobial-resistant pathogens as a growing threat to global public health. Every year, an estimated 1.3 million people die globally due to bacterial AMR [1]. Four of the six leading bacteria that cause AMR-related deaths belong to the ESKAPE pathogens, including *Staphylococcus aureus*, *Klebsiella pneumoniae*, *Acinetobacter baumannii*, and *Pseudomonas aeruginosa* [1]. The impact of AMR extends beyond loss of human life; drug-resistant infections are also far more expensive to treat. According to WHO and World Bank projections, AMR could lead to an additional USD 1 trillion in healthcare costs by 2050, with global GDP losses estimated at USD 1–3.4 trillion per year by 2030 under current trends [2].

Rice used the term “ESKAPE” for the first time in 2008, which is an acronym for a group of six life-threatening pathogens, including both Gram-positive (*S. aureus* and *E. faecium*) and Gram-negative (*A. baumannii*, *K. pneumoniae*, *P. aeruginosa*, and *Enterobacter* spp.) bacteria [3]. They are responsible for the majority of nosocomial infections and capable of escaping the bactericidal activity of antibiotics [4]. Among the ESKAPE pathogens, few groups are more threatening to public health, such as carbapenem-resistant *A. baumannii* (CR-AB), carbapenem-resistant *K. pneumoniae* (CR-KPN), methicillin-resistant *S. aureus* (MRSA), and third-generation cephalosporin-resistant *K. pneumoniae* [5]. While carbapenem-resistant *A. baumannii*, carbapenem-resistant *K. pneumoniae*, and third-generation cephalosporin-resistant *K pneumoniae* each have caused between 50,000 and 100,000 deaths, MRSA alone caused more than 100,000 AMR-related deaths in 2019 [1]. The number is unlikely to fall soon due to the increasing spread of drug resistance in these pathogens and the slow-paced drug development [1].

The spread and pathogenicity of ESKAPE pathogens is partly due to their ability to adhere to tissues and form biofilms on surfaces such as surgical sites, medical implants, and in the lungs of individuals with cystic fibrosis [6]. Biofilm formation enhances survival under hostile conditions and facilitates the spread of antimicrobial resistance, even to initially susceptible strains [7,8]. Within a biofilm, the extracellular polysaccharide matrix and slow-growing ‘persister’ cells create both physical and metabolic barriers that protect the bacteria from antibiotics and immune responses [9]. Bacteria in biofilms can exhibit 10–1000-fold greater antibiotic resistance than planktonic cells [9]. Such infections are notoriously difficult to eradicate. Biofilms on indwelling devices (e.g., catheters, prostheses) or in tissues (e.g., chronic wounds, lungs) lead to persistent, relapsing infections [10]. For instance, *P. aeruginosa* biofilms in cystic fibrosis airways cause chronic pneumonia that is essentially incurable [10]. Infections involving biofilms typically require much longer, more aggressive treatment (and often surgical intervention), driving up healthcare costs and prolonging patient illness. Therefore, the ability of ESKAPE pathogens to form biofilms can complicate both the prevention and treatment of these infections.

The ESKAPE pathogens disproportionately affect vulnerable populations, including immunocompromised individuals, hospitalized patients, and those with underlying illnesses. With rising antimicrobial resistance, these infections can lead to severe, life-threatening complications, extended hospital stays, increased morbidity and mortality rates, and a substantial economic burden on healthcare systems [11,12,13]. This problem poses a greater challenge to low- and middle-income countries (LMIC) due to their often-underdeveloped healthcare systems and limited resources [14]. Moreover, only a few data on the prevalence of drug-resistant ESKAPE pathogens exist in these countries.

Similarly to other LMICs, Bangladesh’s healthcare system is increasingly challenged by the spread of ESKAPE pathogens, some of which have been recognized as the causal agents of emerging infectious diseases in the country [15,16]. Although there has been a recent surge in studies on ESKAPE pathogens in Bangladesh, most of these approaches focus on individual ESKAPE pathogens rather than addressing them collectively. A comprehensive study on the ESKAPE pathogens in Bangladesh is needed to identify the pathogens that evade antibiotic treatment more efficiently than others and require more attention from public health officials. The study was thus designed to explore the current state of five ESKAPE pathogens (*S. aureus*, *E. faecium*, *K. pneumoniae*, *A. baumannii*, and *P. aeruginosa*) in Bangladesh. We compared the drug resistance profiles among these pathogens to identify which ones demonstrate higher levels of multi-drug resistance and pose a significant public health risk. The formation of biofilm by these ESKAPE pathogens may also make the treatment option difficult as the formation of biofilm makes the antibiotics less accessible. Hence, the potential role of biofilm on drug resistance was also investigated in the present study.

## 2. Results

### 2.1. Sample Prevalence

A total of 165 isolates (35 *A. baumannii*, 35 *K. pneumoniae*, 35 *P. aeruginosa*, 30 *S. aureus*, 30 *E. faecium*) of ESKAPE pathogens were collected. The isolates were mostly from urine (29.1%), followed by wound swab (25.45%), tracheal aspirate (16.97%), blood (9.09%), pus (8.5%), sputum (8.5%), cerebrospinal fluid (1.2%), bile (0.6%), and pleural fluid (0.6%) (Figure 1). The sample size was similar for males (n = 77) and females (n = 88). There was no significant age difference between males and females (*p*-value = 0.21, two-tailed Wilcoxon rank-sum test).

### 2.2. Antimicrobial Resistance Pattern of Gram-Positive Isolates of ESKAPE Pathogens

The antibiogram results of Gram-positive ESKAPE pathogens revealed that *E. faecium* isolates were more resistant to the tested antibiotics than *S. aureus* isolates (Figure 2). All the *S. aureus* were sensitive to linezolid, sulfamethoxazole/trimethoprim, gentamicin, and nitrofurantoin. Whereas 60%, 20%, and 13.33% of the *E. faecium* isolates were resistant to nitrofurantoin, gentamicin, and linezolid, respectively. Resistance to tetracyclines (tetracycline and doxycycline) in *S. aureus* was observed only in 6.67% isolates, whereas 46.67% and 33.33% of the *E. faecium* isolates were resistant to tetracycline and doxycycline, respectively. Chloramphenicol was as effective as quinolones against *S. aureus* with 93.33% of the isolates being sensitive. Chloramphenicol was also effective against *E. faecium* with no resistant isolates. Both *S. aureus* and *E. faecium* have developed resistance to fluoroquinolones (ciprofloxacin and levofloxacin). Ciprofloxacin resistance was observed in approximately 53% of *S. aureus* and 86.67% of *E. faecium* isolates. Similarly, levofloxacin resistance was found in 66.67% of *S. aureus* and 86.67% of *E. faecium* isolates. Ampicillin was only tested against *E. faecium* with 86.67% being resistant. Overall, 50% (n = 30) of the Gram-positive isolates were multi-drug resistant (MDR). The percentage of MDR isolates was higher in *E. faecium* (90%, n = 27) than in *S. aureus* (10%, n = 3). Similarly, the cMARI was significantly lower in *S. aureus* than in *E. faecium* (*p*-value < 0.001) (Figure 3a).

### 2.3. Antimicrobial Resistance Pattern of Gram-Negative Isolates of ESKAPE Pathogens

The antibiogram results of Gram-negative ESKAPE pathogens revealed a higher resistance pattern in *A. baumannii* and *K. pneumoniae* and a relatively lower resistance pattern in *P. aeruginosa* (Figure 2). Against carbapenems, the overall resistance was above 40% (imipenem 42.86%, meropenem 47.62%). Carbapenem resistance was highest in *A. baumannii* (imipenem 77.14%, meropenem 74.29%), followed by *K. pneumoniae* (imipenem 34.29%, meropenem 45.71%) and *P. aeruginosa* (imipenem 17.14%, meropenem 22.86%). Fluoroquinolones were even less effective than carbapenems, with 62.86% isolates being resistant to ciprofloxacin and 53.33% isolates being resistant to levofloxacin. Resistance to fluoroquinolones was similar in *A. baumannii* (ciprofloxacin 77.14%, levofloxacin 68.58%) and *K. pneumoniae* (ciprofloxacin 80%, levofloxacin 62.86%), and was lower in *P. aeruginosa* (ciprofloxacin 31.43%, levofloxacin 28.57%).

Resistance to aminoglycosides was above 50% (gentamicin 50.5%, amikacin 57.14%) in the isolates. Resistance to aminoglycosides was similar in *A. baumannii* (gentamicin 62.86%, amikacin 60%) and *K. pneumoniae* (gentamicin 54.29%, amikacin 77.14%) and was lower in *P. aeruginosa* (gentamicin 34.29%, amikacin 34.29%). Similar to aminoglycosides, resistance to cephalosporins (ceftazidime and cefepime) was around 50% (ceftazidime 56.19%, cefepime 50.47%). Resistance was slightly higher in *A. baumannii* (ceftazidime 68.57%, cefepime 77.14%), followed by *K. pneumoniae* (ceftazidime 68.57%, cefepime 48.57%) and *P. aeruginosa* (ceftazidime 31.43%, cefepime 25.71%). Resistance to β-lactam inhibitor (piperacillin/tazobactam) was similar in *A. baumannii* (74.29%) and *K. pneumoniae* (77.14%) and was lower in *P. aeruginosa* (28.57%).

Tetracyclines (tetracycline, doxycycline) and trimethoprim/sulfamethoxazole were tested against only *A. baumannii* and *K. pneumoniae* as recommended by CLSI. Resistance to tetracycline was around 50% (54.3% in *A. baumannii*, 48.6% in *K. pneumoniae*), whereas resistance to doxycycline was 45.71% in both *A. baumannii* and *K. pneumoniae*. A higher percentage of *K. pneumoniae* isolates (60%) were resistant to trimethoprim/sulfamethoxazole compared to *A. baumannii* isolates (54.3%). Aztreonam was tested against *K. pneumoniae* and *P. aeruginosa* as recommended by CLSI. Similar to other antibiotics, resistance to aztreonam was also lower in *P. aeruginosa* (40%) than in *K. pneumoniae* (54.29%)

Overall, 31.4% (n = 33) of isolates were resistant to colistin. The colistin resistance was highest in *K. pneumoniae* (42.86%), followed by *A. baumannii* (25.71%) and *P. aeruginosa* (25.71%) (Figure 2).

A higher percentage of Gram-negative isolates (66.67%, n = 70) were characterized as MDR than Gram-positive isolates (50%, n = 30). Among Gram-negative isolates, the percentage of MDR isolates was nearly similar in *A. baumannii* (77.14%, n = 27) and *K. pneumoniae* (82.86%, n = 29) and was lower in *P. aeruginosa* (40%, n = 14). Similarly, the cMARI difference between *A. baumannii* and *K. pneumoniae* was non-significant (*p*-value = 1.00), whereas cMARI relative to *A. baumannii* (*p*-value < 0.001) and *K. pneumoniae* (*p*-value = 0.003) was significantly lower in *P. aeruginosa*, as shown in Figure 3b. Source variation in MARI was evaluated, isolates sampled from tracheal aspirate showed significantly higher MARI than all other sources (*p*-value < 0.05) (Figure 4a).

### 2.4. Biofilm Formation

The assay revealed that a higher percentage of Gram-positive isolates (95%) formed biofilms compared to Gram-negative isolates (85%) (Figure 5). However, there was no significant difference in relative biofilm formation (RBF) between Gram-positive and Gram-negative isolates (*p*-value = 0.682) (Figure 6c). The percentage of strong biofilm-formers was also similar between Gram-positive (16.7%) and Gram-negative isolates (15.2%). Overall, 88.5% of the isolates were biofilm formers, with 15.8% classified as strong, 13.3% as medium, and 59.4% as weak.

Among Gram-negative ESKAPE bacteria, similar biofilm formation percentages were observed in *A. baumannii* (97%) and *K. pneumoniae* (97%) isolates, with a relatively lower percentage in *P. aeruginosa* (60%). Similarly, there was no significant difference in RBF between *A. baumannii* and *K. pneumoniae* (*p*-value = 1.00) (Figure 6a). However, *K. pneumoniae* had the highest percentage of strong biofilm former (20%) among Gram-negative ESKAPE bacteria, followed by *A. baumannii* (8.6%) and *P. aeruginosa* (8.6%). The RBF of *P. aeruginosa* isolates was significantly lower than *A. baumannii* and *K. pneumoniae* (*p*-value < 0.001) (Figure 6a). Biofilm formation percentage was higher in *E. faecium* than in *S. aureus* isolates (Figure 5). All the *E. faecium* isolates were biofilm former (10% strong, 6.67% medium, and 83.33% weak), whereas 90% (23.33% strong, 6.67% medium, and 60% weak) of the *S. aureus* isolates were biofilm former. However, *S. aureus* isolates had significantly higher RBF than *E. faecium* (*p*-value = 0.007) (Figure 6b). Source variation in the RBF was evaluated; only a significant difference was observed between the isolates from tracheal aspirate and urine samples (*p*-value = 0.003) (Figure 6e).

### 2.5. Correlation Between Antibiotic Resistance and Biofilm Formation

There was no significant difference in relative biofilm formation between MDR and non-MDR isolates (*p*-value = 0.396) (Figure 6f). Biofilm groups were reclassified into considerable (strong and medium) biofilm formation and negligible (weak and no-biofilm formers) biofilm formation. No significant difference in MARI was observed within the two groups (*p*-value = 0.114) (Figure 4c). We also evaluated the correlation between biofilm formation (RBF) and non-susceptibility to the antibiotics included in the cMARI calculation using the Wilcoxon rank-sum test. As shown in Figure 7, isolates with non-susceptibility to carbapenems (imipenem: *p*-value = 0.008 and meropenem: *p*-value = 0.001), cephalosporins (ceftazidime: *p*-value = 0.05 and cefepime: *p*-value = 0.006), and piperacillin/tazobactam (*p*-value = 0.012) had significantly higher biofilm-forming capability. No significant difference in biofilm formation was observed in isolates non-susceptible to fluoroquinolones (ciprofloxacin: *p*-value = 0.078 and levofloxacin: *p*-value = 0.083), aminoglycosides (gentamicin: *p*-value = 0.806 and amikacin: *p*-value = 0.744), and colistin (*p*-value = 0.737). The biofilm formation may facilitate the spread of resistance to antibiotics such as carbapenems, cephalosporins, and piperacillin/tazobactam, but it appears to have no significant effect on resistance to other antibiotics like fluoroquinolones, aminoglycosides, and colistin.

### 2.6. Detection of Carbapenemase and Metallo-β-Lactamase Production

Non-susceptibility to carbapenems (imipenem or meropenem) was approximately 64.8% (n = 68) in 105 g negative bacterial isolates. The isolates showing non-susceptibility to any of the carbapenems were selected for testing carbapenemase (mCIM) production and Metallo-β-lactamase (eCIM) production. Only 25 isolates were found to be mCIM positive suggesting carbapenemase production in 23.8% of the Gram-negative isolates. Carbapenemase production was highest in *K. pneumoniae* (34.3%), followed by *A. baumannii* (31.4%) and *P. aeruginosa* (5.7%). The eCIM test result was interpreted only for mCIM-positive isolates. Based on the eCIM test, 45.8% (11 isolates) of the carbapenemase producer were expressing MBL only, whereas the rest were expressing either only SBL or MBL and SBL both. MBL as sole carbapenemase was found to be highest in *K. pneumoniae* (20%), followed by *A. baumanni* (5.7%) and *P. aeruginosa* (5.7%). There was no significant difference in relative biofilm formation (RBF) between carbapenemase producers and non-producers (*p*-value = 0.024) (Figure 6d). However, carbapenemase producers showed significantly higher MARI than non-producers (*p*-value < 0.001), as shown in Figure 4b.

### 2.7. Screening of MRSA

All the isolates of *S. aureus* were screened for methicillin resistance by molecular and phenotypic methods. Based on the phenotypic results, 50% (n = 15) of the isolates were characterized as MRSA. The *S. aureus* isolates were also investigated for the presence of *mecA* and *mecC*. While no isolates were positive for *mecC*, 46.7% (14 out of 30) of the *S. aureus* were *mecA* carriers (see Figure 8 below). The methicillin resistance can significantly be attributed to the presence of *mecA*, as 86.7% of the isolates that showed resistance to cefoxitin were found positive for *mecA*. However, only one isolate carrying *mecA* was cefoxitin-sensitive.

### 2.8. Screening of VRE and VRSA

In the vancomycin MIC test, all the isolates of *S. aureus* were sensitive, while 20% of the *E. faecium* was vancomycin-resistant. The vancomycin resistance was further evaluated by detecting the *vanA* and *vanB* genes. No isolate of *S. aureus* was *vanA* or *vanB* positive further confirming the phenotypic test. However, 13.33% (n = 4) of the *E. faecium* isolates were positive for *vanB*.

### 2.9. Prevalence of Biofilm-Related Genes Compared to Pan-Genomes

The frequency of each biofilm-forming gene in our study was compared to the pan-genomes from the PanX database and listed in Appendix A (Figure 9).

## 3. Discussion

Antimicrobial resistance (AMR) in ESKAPE pathogens is giving rise to increasingly dangerous bugs like carbapenem-resistant *Acinetobacter baumannii* (CR-AB), carbapenem-resistant *Klebsiella pneumoniae* (CR-KPN), and methicillin-resistant *Staphylococcus aureus* (MRSA), which are contributing to a significant number of AMR-related deaths worldwide [1]. Like other countries, these bugs are also causing a major problem in treating nosocomial and community-acquired infections in Bangladesh. Their prevalence and other virulent properties must be thoroughly characterized to address the treatment failure problem and find the pathogen that demands much wider attention. With that intent, an investigation was conducted to determine the prevalence and biofilm formation of MRSA, CR-AB, CR-KPN, and VRE. Out of the 165 clinical isolates investigated in the study, the CR-AB (77.14%) was the most common, followed by CR-KPN (48.57%), MRSA (50%), CR-PA (22.86%), and VRE (20%). The CR-AB, CR-KPN, and CR-PA were determined by resistance to any of the carbapenems (imipenem or meropenem), whereas MRSA and VRE were identified by cefoxitin disc diffusion assay and MIC of vancomycin, respectively.

In the treatment of Gram-negative bacterial infection, carbapenems are considered last-resort antibiotics. The higher frequency of CR-AB and CR-KPN in our study suggests carbapenems might not be the option in treating Gram-negative bacterial infection caused particularly by *A. baumannii* and *K. pneumoniae*. The hierarchy of carbapenem resistance in the Gram-negative ESKAPE pathogens has also been reflected in the study by Khawla et al. [17]. In Bangladesh, CR-AB and CR-KPN were reported to be even more prevalent in other studies [18,19]. Carbapenem inactivation by producing carbapenemase (Serine carbapenemases or Metallo-β-lactamases) is one of the modes of carbapenemase resistance in Gram-negative isolates [20]. The carbapenemase production in this study was investigated following the mCIM (modified Carbapenemase Inactivation method) test. The mCIM test has better sensitivity than other conventional methods, as demonstrated by Abed et al. [21]. The mCIM test result suggests that 36.77% of the carbapenem-resistant isolates were carbapenemase producers with *K. pneumoniae* being the most common. We also compared the antibiotic-resistant pattern between carbapenemase producers and non-producers. The carbapenemase producers had higher MARI than non-producers suggesting that they are more likely to acquire antibiotic resistance.

Other than CR-AB and CR-KPN, MRSA has been the leading cause of AMR-related deaths. MRSA is virtually resistant to all beta-lactam antibiotics [22]. In Bangladesh, the MRSA percentage was around 65% in 2018, which has increased rapidly from 37.2% in 1998 [23,24,25]. A slightly lower percentage (50%) was found in this study, which might not be a threat considering the high level of sensitivity to other antibiotics (linezolid, sulfamethoxazole/trimethoprim, gentamicin, and nitrofurantoin). Vancomycin-resistant *E. faecium* (VRE) isolates are also a significant public health threat as VRE infections are difficult to treat because of the limited number of effective antibiotics [26]. Although the percentage of VRE (20%) was even lower than that of MRSA, the *E. faecium* isolates showed a relatively high resistance level, as indicated by a higher MDR percentage and high cMARI values (Figure 3).

Among Gram-negative ESKAPE pathogens, *A. baumannii* and *K. pneumoniae* both showed higher antibiotic resistance and greater biofilm-forming capacity than *P. aeruginosa*. CR-AB and CR-KPN, therefore, might be more persistent in the environment. It seems reasonable to consider a correlation between biofilm formation and antibiotic resistance. Several studies previously reported a relationship between biofilm-forming and drug-resistant isolates [27,28]. Taking that into account, we also explored the potential role of biofilm in promoting drug resistance in Gram-negative ESKAPE pathogens. Although we found no significant association between biofilm formation and overall antibiotic resistance patterns, isolates showing non-susceptibility to some antibiotics (carbapenems, cephalosporins, and piperacillin/tazobactam) demonstrated a significant correlation with increased biofilm formation.

The biofilm formation by the ESKAPE pathogens was also investigated genotypically by PCR of species-specific biofilm-forming genes. The frequency of each gene was then compared to the corresponding frequency within the pan-genome from PanX of its respective species. The pan-genome of a bacterium represents the entire set of genes present in that bacterial population. By comparing with a pan-genome, we can analyze the prevalence of a specific gene of a sub-population relative to the population. In *K. pneumoniae* isolates, the percentage of *mrkA* (34.29%), *mrkD* (20%), and *treC* (71.43%) was considerably lower than its pan-genome (93.6%, 97.4%, and 99.6%, respectively). The *mrkA* and *mrkD* coded proteins are essential for rapid biofilm formation in *K. pneumoniae*. The *mrkA* coded MrkA is the main fimbrial subunit, whereas MrkD with adhesion characteristics is located at the tip of the fimbriae and determines the specificity of fimbrial binding [29]. The lower prevalence of *mrkA* and *mrkD* in *K. pneumoniae* may indicate that these isolates rely on non-fimbrial mechanisms for colonization, possibly affecting their ability to establish persistent infections on medical devices or epithelial surfaces. This could influence treatment strategies or the risk of chronic infections, especially in catheterized or ICU patients. The gene *treC* affects biofilm formation by modulating the production of capsular polysaccharide (CPS), a major component of the biofilm matrix [30]. Their slightly lower percentage suggests that the studied population might have lower CPS forming ability.

In *S. aureus*, the percentage of *cna* (100%) was significantly higher than its pan-genome (33%). The *cna* codes a protein that allows the bacteria to adhere to and colonize host tissues, particularly those rich in collagen [31]. The high prevalence of *cna* in *S. aureus* is particularly concerning in the context of bone and joint infections, such as osteomyelitis or prosthetic joint infections. Its universal presence in our isolates highlights the need for enhanced surveillance of collagen-rich tissue infections in hospital settings. However, the percentage of *fnbA* (67.9%) was significantly lower than the pan-genome (96.4%). The *fnbA* encodes fibronectin, a protein found in the extracellular matrix that facilitates the initial attachment of *S. aureus* to various surfaces, such as epithelial cells, endothelial cells, and medical implants [32]. In *E. faecium*, the percentage of *gelE* (40%) was significantly lower than its pan-genome (60.6%). The *gelE* codes for a protein called gelatinase, which is a type of protease enzyme [33]. The lower presence of *gelE* in *E. faecium* could suggest reduced virulence, but the high resistance rates seen in this species indicate that antibiotic treatment remains a significant challenge even if invasiveness is moderate. Continuous genomic monitoring is necessary to detect shifts in virulence gene distribution. The prevalence of other genes was not significantly different from the pattern of pan-genomes.

Overall, the gene-level insights gained in this study underscore the diversity and adaptability of ESKAPE pathogens in clinical environments. Understanding the genomic makeup, especially in terms of virulence and biofilm-associated genes, not only informs treatment decisions but also aids in refining risk assessments for hospital-acquired infections. In low- and middle-income countries like Bangladesh, where routine sequencing is limited, targeted gene surveillance may serve as a cost-effective approach to tracking pathogen evolution and improving infection control policies. Nevertheless, this study has limitations. The reliance on conventional PCR rather than whole-genome sequencing restricted our ability to detect novel or divergent gene variants, and the pan-genome comparisons were limited by relatively small reference datasets (n ≤ 500). Despite these constraints, our findings offer a valuable baseline for targeted surveillance and intervention strategies against high-risk nosocomial pathogens.

## 4. Materials and Methods

### 4.1. Sample Collection and Detection

A total of 165 isolates (35 *A. baumannii*, 35 *K. pneumoniae*, 35 *P. aeruginosa*, 30 *S. aureus*, 30 *E. faecium*) of ESKAPE pathogens were collected from 9 different specimens (tracheal aspirate, wound swab, urine, sputum, cerebrospinal fluid, pus, blood, pleural fluid, and bile) from admitted and outpatient cases between January 2023 and January 2024, with a focus on the statistical comparison of drug resistance and biofilm formation. The sample size was determined using the Statskingdom website (statskingdom.com). For Gram-negative ESKAPE pathogens (*A. baumannii*, *K. pneumonia*, and *P. aerurginosa*), sample size was determined to be 105 isolates, 35 in each, by considering a 95% confidence level, large effect, and the ANOVA test. For Gram-positive ESKAPE pathogens (*S. aureus* and *E. faecium*), sample size was determined to be 60 isolates, 30 in each, by considering a 95% confidence level, large effect, and the *t*-test. This study was approved by the Subcommittee on Ethical Clearance of Research Proposals Involving Human Participants, Faculty of Biological Sciences, University of Dhaka (Approval No. 290/Biol.Scs/).

For the initial isolation and presumptive identification, we employed selective and differential media tailored to each ESKAPE pathogen. CHROMagar Acinetobacter was used to isolate *A. baumannii*, allowing its identification based on characteristic colony color and morphology. MacConkey agar was used for *K. pneumoniae*, where non-motile, lactose-fermenting colonies appear large, mucoid, and pink. Cetrimide agar was used to isolate *P. aeruginosa*, which selectively promotes its growth and often exhibits a characteristic greenish pigmentation and grape-like odor. Mannitol salt agar (MSA) was used for *S. aureus*, where colonies ferment mannitol and appear yellow against a pink background. Enterococcus selective agar, such as bile esculin azide agar, was used for isolating *E. faecium*, which produces black colonies due to esculin hydrolysis. PCR is a widely used molecular biology technique that allows the selective amplification of specific DNA sequences, making it possible to detect and identify bacterial species with high precision. It involves repeated cycles of denaturation (separating DNA strands), annealing (binding of specific primers to target DNA), and extension (synthesis of new DNA strands), resulting in the exponential amplification of the target gene. For species-level identification, we targeted the following specific genes: (*nuc* for *S. aureus*, *blaOXA-51* for *A. baumanii*, *ddl* for *E. faecium*, ITS region for *K. pneumoniae*, *oprL* for *P. aeruginosa*) [34,35,36,37,38]. The primers and PCR conditions for amplifying species-specific genes are listed in Appendix A. Each PCR reaction was performed in a total volume of 25 µL, comprising 12.5 µL of 2× PCR master mix (containing Taq DNA polymerase, deoxynucleotide triphosphates [dNTPs], and reaction buffer), 1 µL each of forward and reverse primers (10 µM), 2 µL of template DNA, and 8.5 µL of nuclease-free water. The genomic DNA from the isolates was obtained using the QIAamp DNA Mini Kit (QIAGEN, Hilden, Germany) following the manufacturer’s guidelines. The DNA concentration was determined with the Colibri Microvolume Spectrometer (Titertek-Berthold, Berthold Detection Systems GmbH, Bleichstrasse, Pforzheim, Germany), while the integrity of the DNA was assessed through electrophoresis on 0.8% (*w*/*v*) agarose gels. These DNA samples were preserved at −20 °C until they were used for PCR assays.

### 4.2. Antimicrobial Susceptibility Testing (AST)

Antimicrobial susceptibility testing (AST) was performed using the Kirby–Bauer disk diffusion method on Mueller–Hinton agar, following Clinical and Laboratory Standards Institute (CLSI) guidelines [39]. In this method, bacterial suspensions were prepared to match a 0.5 McFarland turbidity standard and uniformly spread across the surface of Mueller–Hinton agar plates using sterile swabs. Antibiotic-impregnated paper disks were then placed on the inoculated plates, which were subsequently incubated at 37 °C for 16–18 h. After incubation, the diameter of the zones of inhibition surrounding each antibiotic disk was measured in millimeters using a ruler or caliper. These measurements were interpreted as sensitive, intermediate, or resistant based on the standardized breakpoints provided in CLSI M100-ED34:2024 [40]. The antibiotics used in the susceptibility testing are listed in Table 1. Quality control was performed using *S. aureus* ATCC 25923, *P. aeruginosa* ATCC 27853, and *E. coli* ATCC 25922. The tested isolates were considered multi-drug resistant if they showed resistance to at least one antibiotic in three or more different antimicrobial categories [41].

### 4.3. Determination of Multiple Antibiotic Resistance Index

Based on the results of antibiotic susceptibility testing, Multiple Antibiotic Resistance Index (MARI) was estimated. MARI is a well-established epidemiological indicator used to assess the risk level of environments where bacteria are exposed to antibiotics. The MARI ranges from 0 (completely susceptible to all tested antibiotics) to 1 (resistant to all tested antibiotics). A high MARI value suggests exposure to high-risk sources of antibiotic use, such as hospital settings, where selective pressure is intense. To calculate MARI for each isolate, the following formula was used:

MARI = Number of antibiotics to which the isolate is resistant/Number of antibiotics to which the isolate is resistant [42].

However, since different species were tested against different antibiotic panels according to CLSI guidelines, MARI values could not be directly compared across species. To enable reliable cross-species comparisons, we introduced a standardized index referred to as common Multiple Antibiotic Resistance Index (cMARI). This index was calculated using only the antibiotics that were commonly tested within each group of pathogens. For Gram-negative ESKAPE pathogens (*A. baumannii*, *P. aeruginosa*, and *K. pneumoniae*), cMARI was calculated using resistance results to the following 10 common antibiotics: gentamicin, amikacin, ciprofloxacin, levofloxacin, ceftazidime, cefepime, imipenem, meropenem, piperacillin/tazobactam, and colistin. For Gram-positive ESKAPE pathogens (*S. aureus* and *E. faecium*), cMARI was based on nine common antibiotics: ciprofloxacin, levofloxacin, tetracycline, doxycycline, gentamicin, linezolid, vancomycin, chloramphenicol, and nitrofurantoin. This approach ensured consistent and meaningful comparison of resistance levels within Gram-positive and within Gram-negative groups, while minimizing variability caused by differing antibiotic panels.

### 4.4. Determination of Colistin Resistance by MIC

Colistin resistance in Gram-negative ESKAPE pathogens was determined by the microtiter plate broth dilution method as recommended by CLSI (CLSI M100-ED34:2024). In brief, overnight bacteria cultured in LB medium were diluted at 1:100 in fresh LB and cultured at 37 °C with shaking at 200 rpm to an optical density at 600 nm of 0.5. Then, 10 μL of 0.1 OD was added into each well of a 96-well microtiter polystyrene tray with 100 μL LB of a series of 2-fold dilutions of colistin. The mixtures were incubated at 37 °C for 24 h. Growth ≤ 4 μg/mL was considered colistin-resistant, and growth below 2 μg/mL was considered colistin-sensitive (CLSI M100-ED34:2024).

### 4.5. Detection of Methicillin-Resistant Staphylococcus aureus (MRSA)

*S. aureus* isolates that test positive for *mecA*, *mecC*, or PBP2a or show phenotypic resistance to cefoxitin are considered methicillin-resistant *S. aureus* (MRSA). In this study, all the isolates of *S. aureus* were screened for methicillin resistance by molecular and phenotypic methods. In the phenotypic method, a disc diffusion assay was performed using the cefoxitin disc (30 µg) following CLSI guidelines. In the molecular method, genes responsible for methicillin resistance, namely, *mecA* and *mecC*, were screened by polymerase chain reaction. The primers and PCR conditions for *mecA* and *mecC* are listed in Appendix A [43]. Each PCR reaction was performed as described in Section 4.1.

### 4.6. Screening of VRE and VRSA

All *S. aureus* and *E. faecium* isolates were screened for vancomycin resistance by phenotypic and molecular methods. According to CLSI guidelines, only the MIC test is suggested for testing susceptibility to vancomycin in *S. aureus*. Consequently, the MIC test was used to determine vancomycin resistance in both *S. aureus* and *E. faecium*, in the same manner as it was applied to colistin. In molecular methods, the presence of *vanA* and *vanB* genes was detected by PCR using primers listed in Appendix A [44]. Each PCR reaction was performed as described in Section 4.1.

### 4.7. Detection of Carbapenemase and Metallo-β-Lactamase Production

Carbapenemase production of carbapenem-resistant Gram-negative bacteria was determined by the Modified Carbapenem Inactivation Method (mCIM), whereas the production of Metallo-β-lactamase (MBL) was parallelly determined by EDTA-Modified Carbapenem Inactivation Method (eCIM) [45,46]. In the mCIM test, 2 mL of trypticase soy broths (TSB) were inoculated with a 10 μL loopful of carbapenem-resistant isolates, vortexed, and a 10 μg meropenem disk (Oxoid, Hampshire, UK) was placed into the mixture. Following 4 h of incubation, the meropenem disk was removed from the vial and placed on a Mueller–Hinton agar (MHA) plate lawned with a 0.5 McFarland suspension of *Escherichia coli* ATCC 25922. After 24 h of incubation, the results were interpreted. The eCIM test was performed simultaneously with the addition of 20 μL of 0.5 M EDTA in 2 mL TSB before inoculation.

In the mCIM test, a zone diameter of 6–15 mm or the presence of pinpoint colonies within a 16–18 mm zone was considered carbapenemase-positive, whereas a zone diameter ≥ 19 mm (clear zone) was considered carbapenemase-negative. The zone diameter between 16 and 18 mm without any colonies was considered inconclusive and a repeat test was carried out (CLSI M100-ED34:2024). The result of the eCIM test was analyzed only for carbapenemase-positive isolates. The eCIM test was interpreted as positive (detection of Metallo-β-lactamase) if the zone of inhibition diameter increased by 5 mm or more compared to the mCIM result and was considered negative if the zone of inhibition diameter was 4 mm or less (CLSI M100-ED34:2024). A positive mCIM result suggests the sole expression of Metallo-β-lactamase (MBL) as carbapenemase, whereas a negative result suggests either the expression of Serine-β-lactamase (SBL) or the expression of both SBL and MBL [46].

### 4.8. Phenotypic Biofilm Formation Assay

The biofilm formation ability of ESKAPE pathogens was determined by microtiter dish biofilm formation assay using 96-well polystyrene plates [47]. Samples were cultured in Luria broth (LB) medium at 37 °C for 24 h with subsequent 1:100 dilution and inoculation into microplate wells (triplicates per isolate). After overnight incubation, liquid and unattached cells were discarded, and wells were gently washed with sterile water, air-dried, and stained with 0.1% crystal violet. Following incubation and rinsing, 33% acetic acid was added to solubilize the dye-cell complex, and absorbance at 560 nm was measured using a plate reader with a 33% acetic acid in water blank.

Optical Density Cut-off (ODc) is the threshold value of optical density that differentiates between test results that are considered significant or positive from those that are not (negative). The optical density cut-off value (ODc) was set as three standard deviations (SD) above the mean of the optical density (OD) of the negative control as shown in the following formula: ODc = average OD of negative control + (3 × SD of negative control). The optical density (OD) was determined by calculating the mean of all the replicates. The results were summarized into four categories based on their optical densities: (1) strong biofilm producer (4 × ODc < OD); (2) medium biofilm producer (2 × ODc < OD ≤ 4 × ODc); (3) weak biofilm producer (ODc < OD ≤ 2 × ODc); and (4) non-biofilm producer (OD ≤ ODc) [48]. The ratio of optical density (OD) to optical density cut-off value (ODc) was used to compare biofilm-forming ability among the ESKAPE pathogens and was referred to as relative biofilm formation (RBF).

### 4.9. Detection and Comparison of Biofilm-Forming Genes

Biofilm forming genes for *S. aureus* (clfA, clfB, fnbpA, cna, isdA, isdB, sdrD , sdrE, and bbp), *E. faecium* (hyl, esp, asa1, gelE, acm, and efaA), *A. baumannii* (csuE, pgaB, ompA, bap, and espA), *P. aeruginosa* (algD, pela, and pslA), and *K. pneumoniae* (fimH, luxS, treC, mrkA, and mrkD) were screened by polymerase chain reaction [37,49,50,51,52,53,54,55,56,57,58,59,60,61,62,63,64]. The primers and PCR conditions for biofilm-forming genes are listed in Appendix A. Each PCR reaction was performed as described in Section 4.1. The prevalence of the listed biofilm-forming genes was then compared to the pan-genomes of each specific bacteria from PanX (https://pangenome.org/). The website offers an interactive visualization and dynamic exploration of bacterial pan-genomes [65]. The absence of a gene from the pan-genome does not necessarily indicate their total absence in the bacteria. Pan-genome constructed from a low sample number, the incomplete annotation of bacterial genomes, or even the characterization as hypothetical may miss the gene of interest. Therefore, only the genes present in the PanX database specific to a bacterium were compared to our study.

### 4.10. Statistical Analysis

All statistical analyses were conducted using R Studio (version 4.2.3), a widely used statistical computing environment. For inferential statistics, non-parametric tests were selected due to the non-normal distribution of the data, as confirmed by preliminary assessments using the Shapiro–Wilk test and visual inspection of histograms and Q-Q plots. The Wilcoxon rank-sum test was used for comparing two independent groups (e.g., age differences between male and female patients). When comparing more than two groups, the Kruskal–Wallis test was applied, followed by Dunn’s post hoc test with Bonferroni correction to identify pairwise differences. In all cases, a *p*-value of <0.05 was considered statistically significant. All statistical analyses and generation of figures were completed using R Studio (version 4.2.3).

## 5. Conclusions

This study comprehensively evaluated the antibiotic resistance patterns, biofilm-forming capabilities, and genotypic characteristics of five ESKAPE pathogens isolated from clinical settings in Bangladesh. We found alarmingly high rates of MDR, particularly among *A. baumannii* and *K. pneumoniae*, with over 75% of isolates showing resistance to last-resort antibiotics such as carbapenems. *P. aeruginosa* exhibited relatively lower resistance levels. Among Gram-positive isolates, *E. faecium* showed greater resistance and higher cMARI values than *S. aureus*, despite a lower prevalence of vancomycin resistance. Biofilm formation was common among all ESKAPE pathogens, with over 85% of isolates identified as biofilm formers. Notably, isolates resistant to carbapenems, cephalosporins, and piperacillin/tazobactam displayed significantly higher biofilm-forming capacity, suggesting a potential role of biofilms in facilitating resistance to these critical antibiotics. These findings meet the research objectives by identifying key resistance trends, establishing a framework for comparing antimicrobial resistance using cMARI, and evaluating both phenotypic and genotypic aspects of biofilm formation. These results underscore the urgent need for improved infection control, surveillance, and antimicrobial stewardship, particularly in low- and middle-income countries. This work contributes to public health efforts by highlighting specific pathogens and resistance mechanisms that warrant immediate attention for targeted interventions and further genomic investigations.

## Figures and Tables

**Figure 1 antibiotics-14-00842-f001:**
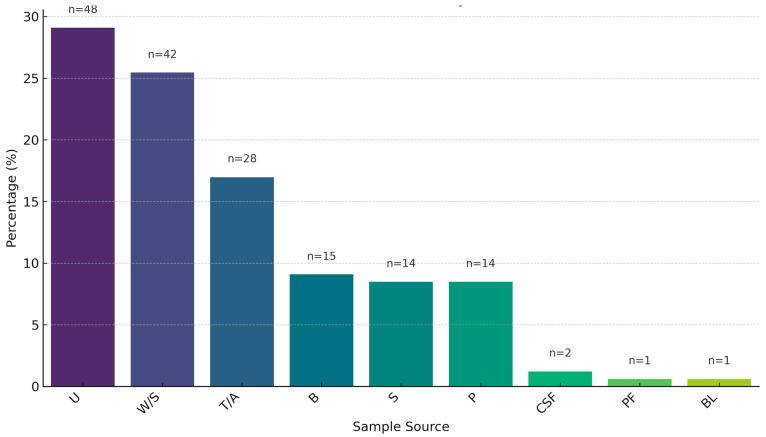
Prevalence of ESKAPE pathogen isolates by clinical sample source (U = Urine, W/S = Wound Swab, TA = Tracheal Aspirate, B = Blood, P = Pus, S = Sputum, CSF = Cerebrospinal Fluid, BL = Bile, PF = Pleural Fluid).

**Figure 2 antibiotics-14-00842-f002:**
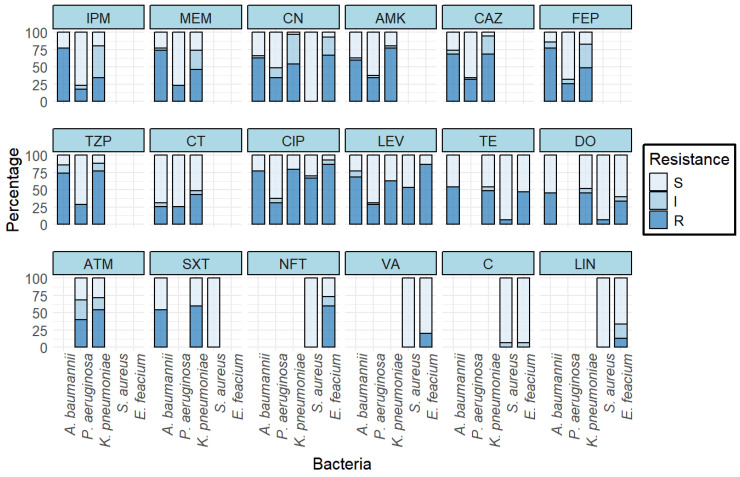
Antibiotic susceptibility profile of ESKAPE pathogens (CN = Gentamicin, AMK = Amikacin, TZP = Piperacillin/tazobactam, IPM = Imipenem, MEM = Meropenem, CAZ = Ceftazidime, FEP = Cefepime, CIP = Ciprofloxacin, LEV = Levofloxacin, SXT = Trimethoprim/sulfamethoxazole, ATM = Aztreonam, C = Chloramphenicol, CT = Colistin, TE = Tetracycline, DO = Doxycycline, VA = Vancomycin, LIN = Linezolid, NFT = Nitrofurantoin, S = Sensitive, I = Intermediate resistance, R = Resistance).

**Figure 3 antibiotics-14-00842-f003:**
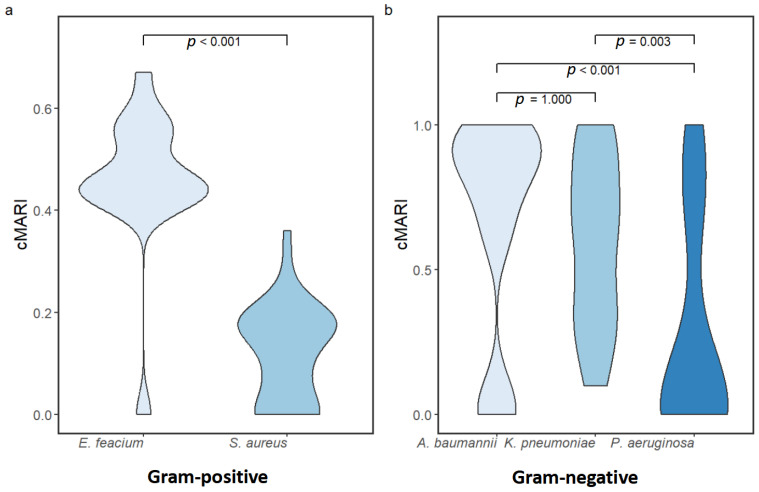
The cMARI (common Multiple Antibiotic Resistance Index) difference within (**a**) Gram-positive and (**b**) Gram-negative ESKAPE pathogens. Difference in the cMARI for Gram-positive isolates were tested by the Wilcoxon rank-sum test (two-sided) and for Gram-negative isolates were tested by the Kruskal–Wallis followed by Dunn post hoc test with Bonferroni correction for multiple comparisons.

**Figure 4 antibiotics-14-00842-f004:**
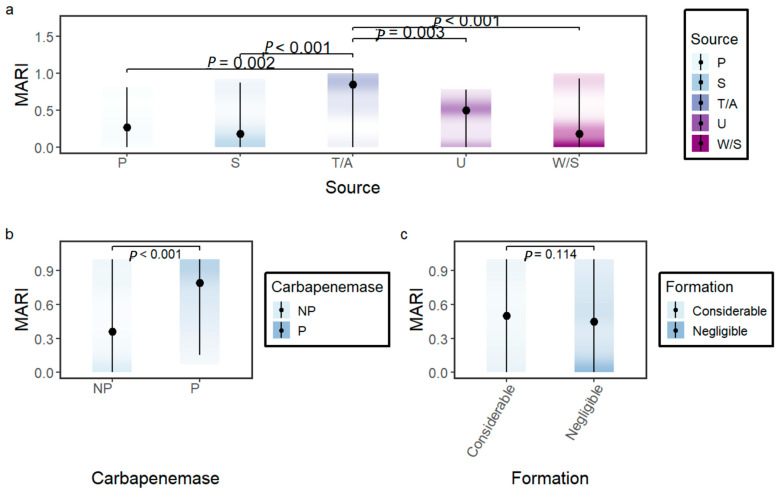
Multiple Antibiotic Resistance Index (MARI) variation within (**a**) sources (P = pus, S = sputum, T/A = tracheal aspirate, W/S = wound swab, U = urine), (**b**) carbapenemase producers (P) and non-producers (NP), and (**c**) biofilm groups: considerable (strong and medium) and negligible (weak and no-biofilm formers). Variation in the MARI values within sources were tested by the Kruskal–Wallis followed by Dunn post hoc test with Bonferroni correction for multiple comparisons. Variation in the MARI values between carbapenemase producers and non-producers and between biofilm groups were tested by the Wilcoxon rank-sum test (two-sided).

**Figure 5 antibiotics-14-00842-f005:**
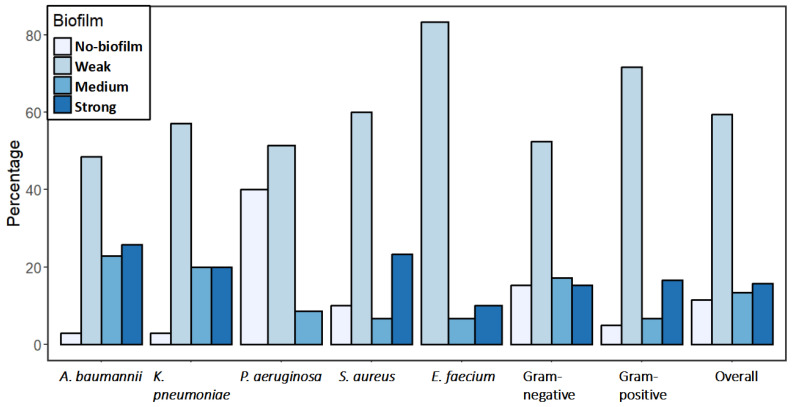
The percentages of biofilm formation (Strong, Medium, Weak, and No-biofilm) among ESKAPE pathogens.

**Figure 6 antibiotics-14-00842-f006:**
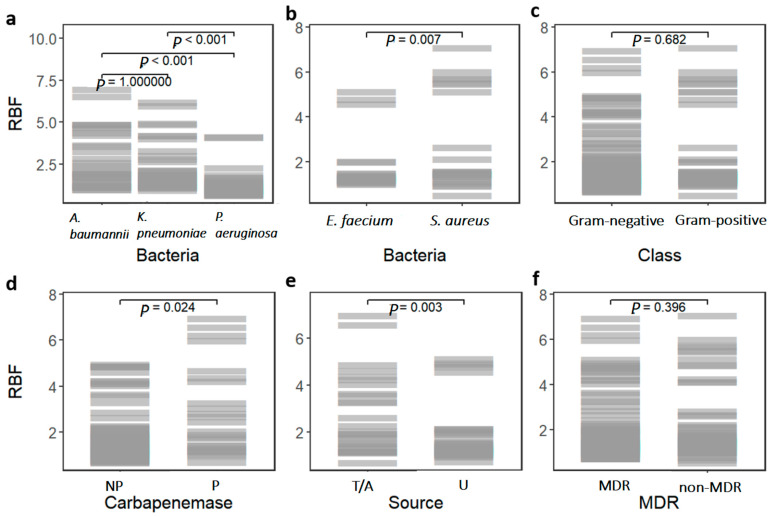
Variation in relative biofilm formation (RBF) in different groups: (**a**) Gram-negative isolates; (**b**) Gram-positive isolates; (**c**) Gram-negative and Gram-positive isolates; (**d**) Carbapenemase producers and non-producers; (**e**) Tracheal aspirate (T/A) and urine (U); (**f**) multi-drug resistant (MDR) and non-MDR. The RBF variation in Gram-negative isolates were tested by the Kruskal–Wallis followed by Dunn post hoc test with Bonferroni correction for multiple comparisons. The rest of the groups were tested by the Wilcoxon rank-sum test (two-sided).

**Figure 7 antibiotics-14-00842-f007:**
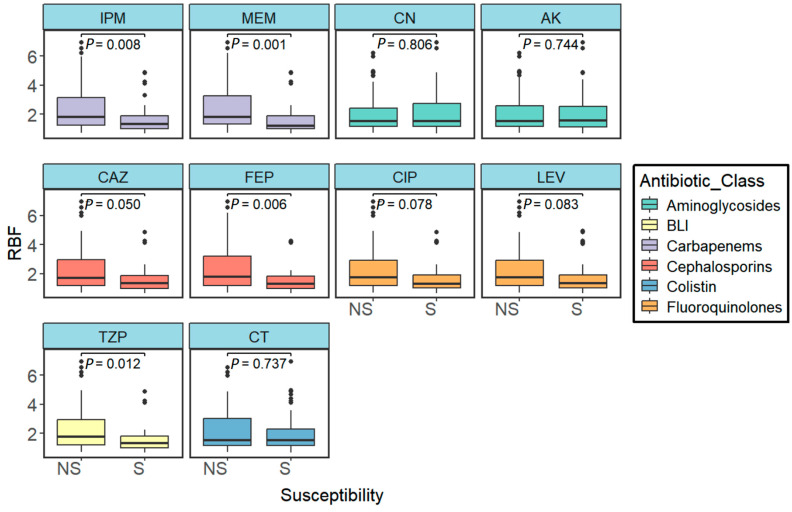
Variation in relative biofilm formation (RBF) in Gram-negative ESKAPE pathogens in susceptible and non-susceptible groups (Imipenem, *p*-value = 0.008; meropenem, *p*-value = 0.001; gentamicin, *p*-value = 0.806; amikacin, *p*-value = 0.744; ceftazidime, *p*-value = 0.05; cefepime, *p*-value = 0.006; ciprofloxacin, *p*-value = 0.078; levofloxacin, *p*-value = 0.083; piperacillin/tazobactam, *p*-value = 0.012; and colistin, *p*-value = 0.737). Differences were tested by the Wilcoxon rank-sum test (two-sided).

**Figure 8 antibiotics-14-00842-f008:**
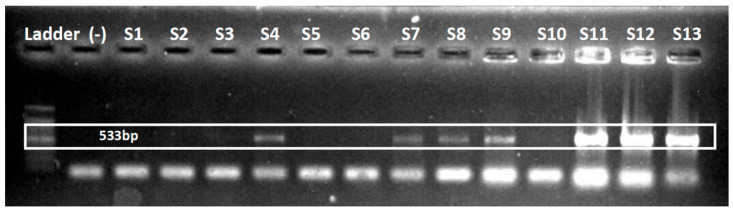
Representative figure of *mecA* gene amplification by PCR.

**Figure 9 antibiotics-14-00842-f009:**
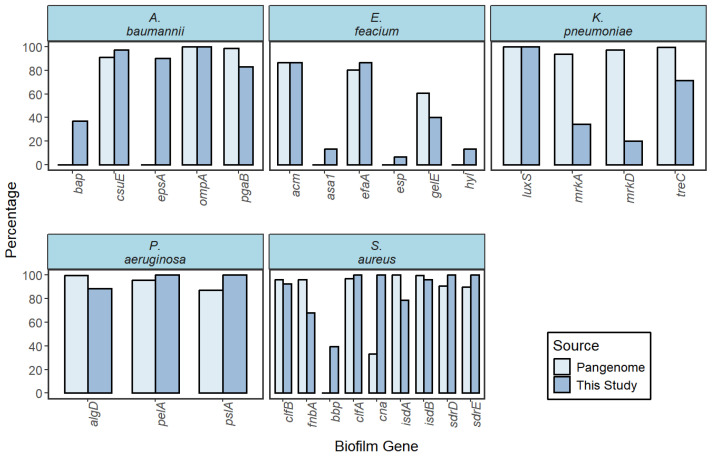
Prevalence of biofilm-related genes in comparison to pan-genomes from PanX. Pan-genome population: *S. aureus* (500), *E. faecium* (127), *K. pneumoniae* (500), *P. aeruginosa* (223), and *A. baumannnii* (205). The absence in the pan-genomes suggests the genes were not recognized during pangenome construction by PanX not their absolute absence.

**Table 1 antibiotics-14-00842-t001:** List of antibiotics used in the study.

Antibiotics Classes	Antibiotics	Gram-Negative			Gram-Positive	
		*A. baumannii*	*P. aeruginosa*	*K. pneumoniae*	*S. aureus*	*E. faecium*
Aminoglycosides	Gentamicin (10 μg)	Yes	Yes	Yes	Yes	Yes
	Amikacin (30 μg)	Yes	Yes	Yes	No	No
Fluoroquinolones	Ciprofloxacin (5 μg)	Yes	Yes	Yes	Yes	Yes
	Levofloxacin (5 μg)	Yes	Yes	Yes	Yes	Yes
Cephalosporins	Ceftazidime (30 μg)	Yes	Yes	Yes	No	No
	Cefepime (30 μg)	Yes	Yes	Yes	No	No
Carbapenems	Imipenem (10 μg)	Yes	Yes	Yes	No	No
	Meropenem (10 μg)	Yes	Yes	Yes	No	No
Penicillins + β-lactamase inhibitors	Piperacillin/tazobactam (100/10 μg)	Yes	Yes	Yes	No	No
Polymyxins	Colistin (MIC)	Yes	Yes	Yes	No	No
Tetracyclines	Tetracycline (30 μg)	Yes	No	Yes	Yes	Yes
	Doxycycline (30 μg)	Yes	No	Yes	Yes	Yes
Monobactams	Aztreonam (30 μg)	No	Yes	Yes	No	No
Folate pathway inhibitors	Trimethoprim/ Sulfamethoxazole (25 μg)	Yes	No	Yes	Yes	No
Oxazolidinones	Linezolid (30 μg)	No	No	No	Yes	Yes
Glycopeptides	Vancomycin (MIC)	No	No	No	Yes	Yes
Phenicols	Chloramphenicol (30 μg)	No	No	No	Yes	Yes
Nitrofurans	Nitrofurantoin (300 μg)	No	No	No	Yes	Yes
Anti-staphylococcal β-lactams	Cefoxitin (30 μg)	No	No	No	Yes	No
Penicillins	Ampicillin (10 μg)	No	No	No	No	Yes

## Data Availability

The datasets GENERATED for this study can be found in the [MARI] [https://github.com/Tasnimul-Arabi-Anik/MARI/blob/main/Supplementary_file_2.csv (accessed on 30 January 2024)].

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
