# Peer review of "Comparative Analysis of Biofilm Formation and Antibiotic Resistance in Five ESKAPE Pathogen Species from a Tertiary Hospital in Bangladesh"

_antibiotics, 2025, doi:10.3390/antibiotics14080842_

Round 1

Reviewer 1 Report

Comments and Suggestions for Authors

The manuscript exmines the antibiotic resistance of five microganisms, three gram negative (A. baumannii, K. pneumoniae and P. aeruginosa) and two gram positive (S. aureus and E. faecium) bacteria belonging to the ESKEPE group and the ability of these pathogens to form biofilms.

The results of antibiogram performed on the gram positive bacteria S. aureus and E. faecium shoved that the latter was more resistant to all antibiotic molecules than S. aureus. Similarly, the percentage of multi-drug resistant (MDR) and the Common Multiple Antibiotic Resistance (cMARI) were significantly lower in S. aureus than E. faecium isolates.

With regard to gram negative ESKAPE bacteria, the antibiogram result revealed a higher panel of resistance in A. baumannii and K. pneumoniae than P. auruginosa against several antibiotic moleculus. The authors first examined susceptibility to carbepenems, which are considered last-resort antibiotics. Among the three ESKAPE gram negative agents, A. baumannii showed the highest resistance, followed by K. pneumoniae and P. aeruginosa. Differently, the grade of resistance to fluoroquinolones, aminoglycosides and cephalosporins was similar between A. baumannii and K. pneumoniae but higher than that of P. aeruginosa. Finally, only K. pneumoniae was more resistant to colistin. Of all the sample examined (tracheal aspirate, wound swab, urine, sputum, cerebrospinal fluid, pus, blood, pleural fluid and bile), the microorganisms isolated from the tracheal aspirate showed much higher MARI.

Furthermore, in this study, the authors also investigated the possible role of biofilm in determining antibiotic resistance by ESKAPE gram negative bacteria. No correlation was observed between resistance and biofilm formation although incresead resistance to carbapenems, cephalosporins and piperacillin/tazobactam was associated with an increased ability to form biofilm.

In conclusion, the authors'aim was to conduct a study on some ESKAPE pathogens present in a tertiary hospital in Bangladesh to identify microorganisms that are able to evade antibiotic treatment more effectively than others and therefore require more attention from the local public health.

The manuscript is relevant to the fields and scientifically sound, with an appropriate experimental design. The figure and tables are appropriate and correctly show the specific data. The resuls of the study are riproducible based on details provided in the method section. The conclusion are consistent with the evidence and arguments presented appropriately.

The only thing i do not understand is the meaning is that sentence in bold print that reads "Error! Reference source not found".

Overall, I would suggest taking the article.

Author Response

Comment1: The only thing i do not understand is the meaning is that sentence in bold print that reads "Error! Reference source not found.

Response: The problem possibly occurred during the conversion of manuscript to the journal’s layout. We have solved the problem in the revised manuscript.

Reviewer 2 Report

Comments and Suggestions for Authors

Dear authors, below I share observations that will improve the quality of the article. Overall, I believe your article has a significant opportunity for publication, but cannot be published in its current version.

  1. A part of the introduction has no references and says the phrase "[Error! Reference source not found.]", the problem must be reviewed and solved as soon as possible.
  2. In the introduction, they mention important information about antimicrobial resistance in ESKAPE bacteria, but they don't mention the impact that this AMR has on public health in these bacteria, so I recommend explaining this so that the introduction is as clear as possible. Additionally, I believe that the importance of biofilm formation in AMR and its impact on public health should be better explained.
  3. This study does not present results on Enterobacter, so it cannot be considered entirely ESKAPE bacteria. I recommend that the results of this bacteria be analyzed so that they can be fully classified as ESKAPE bacteria.
  4. I do not fully agree that cMARI results should be presented in violin plots, because it can generate confusion in readers who are not familiar with these, I suggest the authors use easier to interpret graphics such as bar charts, heat maps, stacked bars, UpSet plots or scatter plots, this will allow a visualization that allows an adequate understanding of these research results.
  5. In Figure 5, I recommend that the letters in sections a-f be in bold and slightly removed from the graphs, to avoid confusion on the part of readers when viewing the graphs.
  6. I recommend that in the "Results" section, authors create at least one figure showing the results of the PCR for the detection of genes related to antimicrobial resistance and biofilm formation. This will allow readers to properly interpret the results of this currently highly important technique.
  7. The "Discussion" section discusses the presence of genes related to antimicrobial resistance and biofilm formation, but I consider that the discussion is not entirely adequate, so I suggest a more detailed discussion about the presence of these genes in the bacteria isolated from your samples, in addition to delving into the impact of these genes on human infection and their role in public health.
  8. The conclusion could be improved. I recommend briefly including your research findings, how your results could impact human health, and whether they adequately meet the research objectives, to further emphasize your results and their impact on public health.
  9. In section "2.1 Sample Prevalence," I suggest explaining the results further. I recommend creating a graph that allows readers to better visualize these results. A graph indicating which hospital departments each bacteria was found in would also be helpful.
  10. I recommend that the microbiological techniques for identifying ESKAPE bacteria be explained in detail in section "5.1. Sample Collection and Detection." This will allow the reader to fully understand the methodologies used in the research.
  11. In section "5.2. Antimicrobial susceptibility testing (AST)" the Kirby-Bauer method should be explained in more detail, this will allow readers to fully understand this technique.
  12. In section "5.3. Determination of Multiple Antibiotic Resistance Index" it is important to explain in detail the procedure for estimating the Multiple Antibiotic Resistance Index (MARI) and also cMARI, this will allow non-specialist readers to properly understand how these indices are obtained and their importance in research.
  13. I consider it extremely important that the "Materials and Methods" section provide a detailed explanation of how PCR is used to identify genes related to the presence of antimicrobial resistance in bacteria and biofilm-forming genes. This will allow non-specialist readers to understand this highly important technique today.
  14. Section "5.10 Statistical Analysis" is brief. I recommend expanding the description of the data, including the statistical analysis performed, and better explaining the processing used in the research, to make it more complete.

Author Response

Comments 1:

A part of the introduction has no references and says the phrase "[Error! Reference source not found.]", the problem must be reviewed and solved as soon as possible.

  • Thank you for pointing this out. We agree with this comment. The problem possibly occurred during the conversion of manuscript to the journal’s layout. We have solved the problem in the revised manuscript

Comments 2: 

In the introduction, they mention important information about antimicrobial resistance in ESKAPE bacteria, but they don't mention the impact that this AMR has on public health in these bacteria, so I recommend explaining this so that the introduction is as clear as possible. Additionally, I believe that the importance of biofilm formation in AMR and its impact on public health should be better explained.

  • We have addressed the public health impact of antimicrobial resistance (AMR) in ESKAPE pathogensin lines 48–51 and 83–87 of the revised introduction. Additionally, the importance of biofilm formation in contributing to AMR and its implications for public health are discussed in lines 69–80, where we elaborate on how biofilms increase treatment complexity, healthcare costs, and infection persistence.

Comments 3:

This study does not present results on Enterobacter, so it cannot be considered entirely ESKAPE bacteria. I recommend that the results of this bacteria be analyzed so that they can be fully classified as ESKAPE bacteria.

  • We focused exclusively on five ESKAPE ( aureus, E. faecium, K. pneumoniae, A. baumannii, and P. aeruginosa)pathogens because our study aimed to work at species level. Since Enterobacter is a genus that includes a wide range of species, we believe it requires a separate, more detailed study to adequately address its virulence characteristics and resistance mechanisms. Besides, the Enterobacter spp is not so commonly detected in our study area. For clarity and accuracy, we have updated the manuscript title to "Five ESKAPE Pathogens" to reflect the focus of our study.

Comments 4:

 I do not fully agree that cMARI results should be presented in violin plots, because it can generate confusion in readers who are not familiar with these, I suggest the authors use easier to interpret graphics such as bar charts, heat maps, stacked bars, UpSet plots or scatter plots, this will allow a visualization that allows an adequate understanding of these research results.

  • Thank you for your insightful feedback regarding the use of violin plots to present cMARI results. We understand that violin plots may be unfamiliar to some readers and acknowledge your concern. While boxplots are a commonly used alternative and were considered, we opted for violin plots because they provide a more informative visualization of the distribution, density, and potential multimodality of the cMARI values across groups. This was particularly important in our dataset, where subtle distributional differences were critical to interpreting patterns in antimicrobial resistance. Unlike boxplots, violin plots illustrate the full shape of the data distribution, allowing readers to detect nuances that may otherwise be overlooked. 

Comments 5:

 In Figure 5, I recommend that the letters in sections a-f be in bold and slightly removed from the graphs, to avoid confusion on the part of readers when viewing the graphs.

  • Thank you for your helpful suggestion regarding the labeling in Figure 5. As recommended, we have updated the figure to display the section labels (a–f) in bold and repositioned them slightly away from the graphs to improve clarity and avoid confusion. We agree that this modification.

Comments 6: 

I recommend that in the "Results" section, authors create at least one figure showing the results of the PCR for the detection of genes related to antimicrobial resistance and biofilm formation. This will allow readers to properly interpret the results of this currently highly important technique.

  • Thank you for your valuable suggestion. As recommended, we have included a figure illustrating the PCR results for the detection of genes related to antimicrobial resistance. This has been added to the manuscript as Figure 8.

Comments 7: 

The "Discussion" section discusses the presence of genes related to antimicrobial resistance and biofilm formation, but I consider that the discussion is not entirely adequate, so I suggest a more detailed discussion about the presence of these genes in the bacteria isolated from your samples, in addition to delving into the impact of these genes on human infection and their role in public health.

  • Thank you for your valuable feedback. In response, we have provided a more detailed discussion of the biofilm- and antimicrobial resistance-associated genes detected in our isolates. Specifically, we elaborated on the clinical relevance and potential impact of genes in lines 372–376, 382–386, and 391–408.

Comments 8: 

The conclusion could be improved. I recommend briefly including your research findings, how your results could impact human health, and whether they adequately meet the research objectives, to further emphasize your results and their impact on public health.

  • We appreciate your suggestion regarding the conclusion. We have revised the section (lines 410–428) to better summarize our key findings, explicitly link them to their impact on human health, and reflect on how the results align with our research objectives.

Comments 9: 

In section "2.1 Sample Prevalence," I suggest explaining the results further. I recommend creating a graph that allows readers to better visualize these results. A graph indicating which hospital departments each bacteria was found in would also be helpful.

  • Thank you for your valuable feedback. We have added Figure 1, which visually represents the prevalence of the five ESKAPE pathogens by sample source to enhance clarity and aid interpretation. Unfortunately, specific information regarding hospital departments associated with each isolate was not available in the collected dataset.

Comments 10:

I recommend that the microbiological techniques for identifying ESKAPE bacteria be explained in detail in section "5.1. Sample Collection and Detection." This will allow the reader to fully understand the methodologies used in the research.

  • Thank you for your helpful recommendation. We have expanded Section 5.1 (lines 445–454) to provide a more detailed explanation of the microbiological techniques used for the identification of ESKAPE pathogens.

Comments 11:

In section "5.2. Antimicrobial susceptibility testing (AST)" the Kirby-Bauer method should be explained in more detail, this will allow readers to fully understand this technique.

  • Thank you for your valuable feedback. In Section 5.2 (lines 477–482), we have now provided a more detailed explanation of the Kirby–Bauer disk diffusion method. These additions aim to make the antimicrobial susceptibility testing procedure clear and accessible for all readers.

Comments 12:

In section "5.3. Determination of Multiple Antibiotic Resistance Index" it is important to explain in detail the procedure for estimating the Multiple Antibiotic Resistance Index (MARI) and also cMARI, this will allow non-specialist readers to properly understand how these indices are obtained and their importance in research.

  • We appreciate your feedback. Section 5.3 has been thoroughly updated to better explain how the Multiple Antibiotic Resistance Index (MARI) and the common MARI (cMARI) were calculated.

Comments 13:

I consider it extremely important that the "Materials and Methods" section provide a detailed explanation of how PCR is used to identify genes related to the presence of antimicrobial resistance in bacteria and biofilm-forming genes. This will allow non-specialist readers to understand this highly important technique today.

  • Thank you for emphasizing the importance of methodological clarity. In response, we have expanded the description in lines 454–459 and 463–467 to provide a more detailed explanation of how PCR was used to detect antimicrobial resistance genes and biofilm-related genes.

Comments 14:

Section "5.10 Statistical Analysis" is brief. I recommend expanding the description of the data, including the statistical analysis performed, and better explaining the processing used in the research, to make it more complete.

  •  Section 5.10 (lines 606–612) has been expanded to offer a more comprehensive overview of the statistical analysis

Reviewer 3 Report

Comments and Suggestions for Authors

I reviewed  Comparative Analysis of Biofilm Formation and Antibiotic Re- 2
sistance in Clinical ESKAPE Pathogens from a Tertiary Hospital in Bangladesh   an interesting and important paper.

The article is organized as The instructions for authors require, methods are well described, results are clearly as the conclusions.

My only recommandation is to highlight the relationship between ESKAPE pathogens and biofilm formation, the aim of the study, and conclusions.

Author Response

Comment1: My only recommandation is to highlight the relationship between ESKAPE pathogens and biofilm formation, the aim of the study, and conclusions.

Response1: Thank you for your thoughtful recommendation. In response, we have clarified and emphasized the relationship between ESKAPE pathogens and biofilm formation in the revised Introduction (lines 69–80). Additionally, the Conclusion section (lines 410–428) has been revised to better reflect the study’s objectives and highlight the public health significance of our findings.

Reviewer 4 Report

Comments and Suggestions for Authors

Unfortunately the title of this article and the actual research do not correspond. ESKAPE pathogens means - Enterococcus faecium, Staphylococcus aureus, Klebsiella pneumoniae, Acinetobacter baumannii and Enterobacter. The last section was not tackled in this article and therefore I do not recomand this article for publishing. Before resubmitting the article please address this major issue. 

Author Response

Comments1: Unfortunately the title of this article and the actual research do not correspond. ESKAPE pathogens means - Enterococcus faecium, Staphylococcus aureus, Klebsiella pneumoniae, Acinetobacter baumannii and Enterobacter. The last section was not tackled in this article and therefore I do not recomand this article for publishing. Before resubmitting the article please address this major issue. 

Response1: Thank you for your critical observation regarding the use of the term "ESKAPE" in the manuscript title. We agree that the full ESKAPE acronym includes Enterobacter spp., which was not included in our study. However, our study was intentionally focused on five species-level ESKAPE pathogensEnterococcus faecium, Staphylococcus aureus, Klebsiella pneumoniae, Acinetobacter baumannii, and Pseudomonas aeruginosa—due to their higher clinical prevalence and availability of isolates from our study site. 

 Since Enterobacter is a genus that includes a wide range of species, we believe it requires a separate, more detailed study to adequately address its virulence characteristics and resistance mechanisms. Besides, the Enterobacter spp is not so commonly detected in our study area. For clarity and accuracy, we have updated the manuscript title to "Five ESKAPE Pathogens" to reflect the focus of our study.

Round 2

Reviewer 2 Report

Comments and Suggestions for Authors

Dear authors, I believe that all the observations I shared with you have been addressed. Except for two minor observations, which I share with you below, the article is quite complete and of a good quality for publication in the journal. Congratulations. Best regards.

  1. In the abstract, it is important to have sections such as: background, methods, results, and conclusions. This will allow you to delimit the sections so that readers can better understand the information in that section.
  2. On line 67 of the introduction it says "[Error! Reference source not found.]", I suggest being extremely meticulous about this, you must make absolutely sure that the citations are written correctly.

Author Response

Comment1:

In the abstract, it is important to have sections such as: background, methods, results, and conclusions. This will allow you to delimit the sections so that readers can better understand the information in that section.

Response1: 

Thank you for your valuable suggestions. In response, we have revised the abstract to clearly include the sections: Background, Methods, Results, and Conclusions. This structured format will help readers quickly grasp the key aspects of the study.

Comment2:

On line 67 of the introduction it says "[Error! Reference source not found.]", I suggest being extremely meticulous about this, you must make absolutely sure that the citations are written correctly.

Response2:

Regarding the citation issue on line 67, we acknowledge the “[Error! Reference source not found.]” note and agree with your concern. This error likely occurred during the automated formatting or file conversion process of the journal system. We have thoroughly reviewed the manuscript to ensure all references are now properly cited and formatted prior to this resubmission.

Reviewer 4 Report

Comments and Suggestions for Authors

The quality of the paper improved considerably. However:

  • The title in the current form might be misleading - some people might think that there are only 5 samples.
  • Figure 3 - Gram-positive and Gram-negative – capital letter.
  • Figure 5 - Gram-positive and Gram-negative – capital letter and the Legend should be with capital letter too – the part regarding the biofilm.
  • Figure 6 - Gram-positive and Gram-negative – capital

Author Response

Comment1:

  • The title in the current form might be misleading - some people might think that there are only 5 samples.
  • Figure 3 - Gram-positive and Gram-negative – capital letter.
  • Figure 5 - Gram-positive and Gram-negative – capital letter and the Legend should be with capital letter too – the part regarding the biofilm.
  • Figure 6 - Gram-positive and Gram-negative – capital

Response1: Thank you for your helpful suggestions. We have revised the title to clarify that the study focuses on five ESKAPE pathogen species, not five samples. Additionally, the terms Gram-positive and Gram-negative have been capitalized in Figures 3, 5, and 6 as recommended. We have also updated the legend in Figure 5